# MIXTURE-OF-SUPERNETS: IMPROVING WEIGHT-SHARING SUPERNET TRAINING WITH ARCHITECTURE-ROUTED MIXTURE-OF-EXPERTS

## ABSTRACT

Weight-sharing supernet has become a vital component for performance estimation in the state-of-the-art (SOTA) neural architecture search (NAS) frameworks. Although supernet can directly generate different subnetworks without retraining, there is no guarantee for the quality of these subnetworks because of weight sharing. In NLP tasks such as machine translation and pre-trained language modeling, we observe that given the same model architecture, there is a large performance gap between supernet and training from scratch. Hence, supernet cannot be directly used and retraining is necessary after finding the optimal architectures.

In this work, we propose *mixture-of-supernets*, a generalized supernet formulation where mixture-of-experts (MoE) is adopted to enhance the expressive power of the supernet model, with negligible training overhead. In this way, different subnetworks do not share the model weights directly, but do so indirectly through an architecture-based routing mechanism. As a result, model weights of different subnetworks are customized towards their specific architectures and the weight generation is learned by gradient descent. Compared to existing weight-sharing supernet for NLP, our method can minimize the retraining time, greatly improving training efficiency. In addition, the proposed method achieves the SOTA performance in NAS for building fast machine translation models, yielding better latency-BLEU tradeoff compared to HAT, the state-of-the-art NAS for MT. We also achieve the SOTA performance in NAS for building memory-efficient task-agnostic BERT models, outperforming NAS-BERT and AutoDistil in various model sizes.

## 1 INTRODUCTION

Neural architecture search (NAS) can automatically design architectures that achieve high quality on the natural language processing (NLP) task, while satisfying user-defined efficiency (e.g., latency, memory) constraints (Wang et al., 2020a; Xu et al., 2021; 2022a). Most straightforward way of NAS is treating it as the black-box optimization (Zoph et al., 2018; Pham et al., 2018). However, to get the architecture with the best accuracy, different model architectures need to be repeatedly trained and evaluated, which makes it impractical unless the dataset is very small. To overcome this issue, weight sharing is applied between different model architectures (Pham et al., 2018). In this case, *supernet* is constructed as the largest model in the search space, and each architecture is a subnetwork of it. Furthermore, recent works (Cai et al., 2020; Yu et al., 2020) show that with good training strategies, the subnetworks can be directly used for image classification with high performance (e.g., accuracy comparable to training the same architectures from scratch). However, it is more challenging to apply supernet in NLP tasks. In fact, we observed that directly using the subnetworks for NLP tasks can have a large performance gap. This is consistent with the recent NAS works (Wang et al., 2020a; Xu et al., 2021) on NLP, which retrain or finetune the architectures after using supernet to find the architecture candidates. This raises two issues: 1) it is unknown whether the selected architectures are optimal given the existence of this performance gap; 2) repeated training is still needed if we want to get the final accuracy of the Pareto front, i.e., the best models for different efficiency (e.g., model size or inference latency) budgets. In this work, we focus on improving the weight-sharing mechanism among subnetworks to minimize the performance gap.

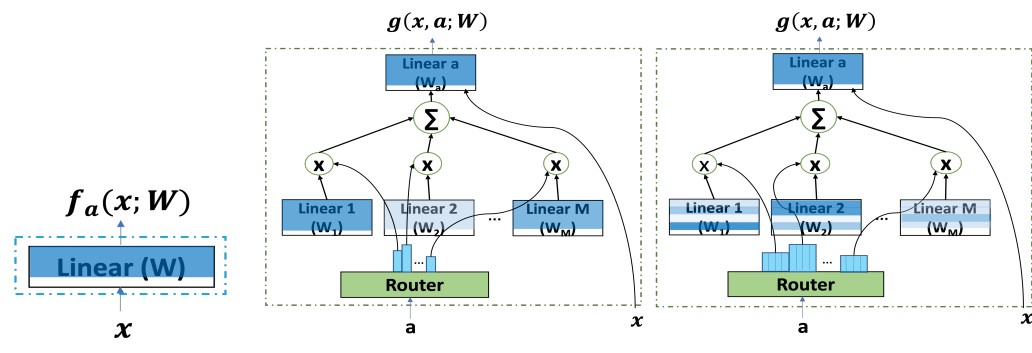

|                    |                                |                                  |
| ------------------ | ------------------------------ | -------------------------------- |
| (a) Standard       | (b) Layer-wise Mixture-of-Supernet | (c) Neuron-wise Mixture-of-Supernet |

Figure 1: Choices of linear layers for supernet training. The length and the height of the 'Linear' blocks correspond to the number of input and output features of the supernet respectively. The highlighted portions in blue color correspond to the architecture-specific weights extracted from the supernet. Different intensities of blue color in the 'Linear' blocks of the mixture-of-supernet correspond to different alignment scores generated by the router.

| Supernet | Weight sharing | Capacity | Overall Time (↓) | Average BLEU (↑) |
| --- | --- | --- | --- | --- |
| HAT (Wang et al., 2020a) | Strict | Single Set | 508 hours | 25.93 |
| Layer-wise MoS | Flexible | Multiple Set | 407 hours (20%) | 27.21 (4.9%) |
| Neuron-wise MoS | Flexible | Multiple Set | **394 hours (22%)** | **27.25 (5.1%)** |

Table 1: Overall time savings and average BLEU improvements of MoS supernets vs. HAT for computing pareto front (latency constraints: 100 ms, 150 ms, 200 ms) for the WMT'14 En-De task. Overall time (single NVIDIA V100 hours) includes supernet training time, search time, and additional training time for the optimal architectures. Average BLEU is the average of BLEU scores of architectures in the pareto front (see Table 5 for individual scores). MoS supernets yield architectures that enjoy better latency-BLEU trade-offs than HAT and have an overall GPU hours (see A.4.10 for breakdown) savings of at least 20% w.r.t. HAT.

Typically, weight-sharing supernet is trained by repeatedly sampling an architecture from the search space and training the architecture-specific weights from the supernet (see Figure 1 (a)). In the standard weight-sharing training (Yu et al., 2020; Cai et al., 2020), the first few output neurons are directly extracted to form a smaller subnetwork, as shown in Figure 1 (a). Such a supernet has limited model capacity, which creates two challenges. First, the supernet enforces a strict notion of weight sharing between architectures, regardless of the difference among these architectures. This leads to the issue of co-adaptation (Bender et al., 2018; Zhao et al., 2021c) and gradient conflict (Gong et al., 2021). For instance, given a 5M-parameters model as a subnetwork of a 90M-parameters model, 5M weights are directly shared in the standard weight-sharing. The optimal shared weights for the 5M model could be non-optimal for the 90M model, since there could be large gradient conflicts in optimizing these two models (Gong et al., 2021). Second, the overall capacity of the architecture allocated by the supernet is limited by the number of parameters of a single DNN, i.e. the largest subnetwork in the search space. However, the number of subnetworks in the search space could be very large (e.g., billions). Using a single set of weights to simultaneously parameterize all of them could be insufficient (Zhao et al., 2021c). Due to these challenges, the gap between the performance of the supernet and the standalone (from scratch) model is usually large (Wang et al., 2020a; Ganesan et al., 2021; Yin et al., 2021), which makes the time consuming retraining step of the optimal architectures mandatory.

To overcome these challenges, we propose a Mixture-of-Supernets (MoS) framework that can perform architecture-specific weight extraction (e.g., allows a smaller architecture to not share some output neurons with a larger architecture) and allocate large capacity to an architecture without being limited by the number of parameters in a single DNN. MoS maintains a set of expert weight matrices and has two variants: *layer-wise MoS* and *neuron-wise MoS*. In layer-wise MoS, architecture-specific weight matrix is constructed based on a weighted combination of expert weight matrices at the level of set of neurons corresponding to an expert weight matrix. On the other hand, neuron-wise MoS constructs the same at the level of an individual neuron in each expert weight matrix. We show the

effectiveness of the proposed NAS method for building efficient task-agnostic BERT (Devlin et al., 2019) models and machine translation (MT) models. For building efficient BERT, our best supernet: (i) closes the gap and improves over SuperShaper (Ganesan et al., 2021) by 0.85 GLUE points, (ii) improves over NAS-BERT (Xu et al., 2021) and AutoDistil (Xu et al., 2022a) in various model sizes ($\leq 50M$ parameters). Compared to HAT (Wang et al., 2020a), our best supernet: (i) reduces the supernet vs. the standalone model gap by 26.5%, (ii) yields a better pareto front for latency-BLEU tradeoff (100 to 200 ms), and (iii) reduces the number of additional steps to close the gap by 39.8%. See Table 1 for a summary of the overall time savings and BLEU improvements of MoS supernets for WMT'14 En-De task. For this task, the supernet training time is 248 hours, while neuron-wise MoS and layer-wise MoS require additional hours of 14 and 18 hours respectively (less than 8% overhead, see A.4.10 for breakdown).

**Main contributions: (1)** We propose a formulation which can generalize weight sharing methods, including direct weight sharing (e.g., once-for-all network (Cai et al., 2020), BigNAS (Yu et al., 2020)) and flexible weight sharing (e.g., few-shot NAS (Zhao et al., 2021a)). This formulation allows us to improve supernet by enhancing the model's expressive power. **(2)** We adopt the idea of MoE to improve the model capability. Specifically, the model's weights are dynamically generated based on the activated subnetwork architecture. After training, this MoE can be converted into equivalent static models. This is because our supernets only depend on the subnetwork architecture, which is fixed after training. **(3)** We conduct comprehensive experiments, demonstrating that our supernets achieve the SOTA NAS results on building efficient task-agnostic BERT and MT models.

## 2 SUPERNET - FUNDAMENTALS

Supernet is a model that employs weight sharing to parameterize weights for millions of architectures. Supernet can provide quick performance predictions for various architectures, which reduces the search cost for NAS significantly. The training objective of the supernet can be formalized as follows. Let $\mathcal{X}_{tr}$ denote the training data distribution. Let $x, y$ denote the training sample and label respectively, i.e., $x, y \sim \mathcal{X}_{tr}$. Let $a_{rand}$ denote an architecture uniformly sampled from the search space $\mathcal{A}$. Let $f_a$ denote the subnetwork with architecture $a$, and $f$ be parameterized by the supernet model weights $W$. Then, the training objective of the supernet can be given by,

$$\min_W \mathbb{E}_{x,y \sim \mathcal{X}_{tr}} \mathbb{E}_{a_{rand} \sim \mathcal{A}}[\mathcal{L}(f_{a_{rand}}(x; W), y)]. \tag{1}$$

The above formulation is known as single path one-shot (SPOS) optimization (Guo et al., 2020) of supernet training. Sandwich training (Yu et al., 2020) is another popular technique for training a supernet, where the largest architecture ($a_{big}$), the smallest architecture ($a_{small}$), and the architecture ($a_{rand}$) uniformly sampled from the search space are jointly optimized. The training objective of the supernet then becomes:

$$\min_W \mathbb{E}_{x,y \sim \mathcal{X}_{tr}}[\mathbb{E}_{a_{rand} \sim \mathcal{A}}[\mathcal{L}(f_{a_{rand}}(x; W), y)] + \mathcal{L}(f_{a_{big}}(x; W), y) + \mathcal{L}(f_{a_{small}}(x; W), y)]. \tag{2}$$

## 3 MIXTURE-OF-SUPERNETS

Existing supernets typically have limited model capacity to extract architecture-specific weights. For simplicity, assume the model function $f_a(x; W)$ is a fully connected layer (output $o = Wx$, omitting bias term for brevity), where $x \in n_{in} \times 1$, $W \in n_{out} \times n_{in}$, and $o \in n_{out} \times 1$. $n_{in}$ and $n_{out}$ correspond to the number of input and output features respectively. Then, the weights ($W_a \in n_{out_a} \times n_{in}$) specific to architecture $a$ with $n_{out_a}$ output features are typically extracted by taking the first $n_{out_a}$ rows[1] (as shown in Figure 1 (a)) from the supernet weight $W$. Assume one samples two architectures ($a$ and $b$) from the search space with the number of output features $n_{out_a}$ and $n_{out_b}$ respectively. Then, the weights corresponding to the architecture with the smallest number of output features will be a subset of those of the other architecture, sharing the first $|n_{out_a} - n_{out_b}|$ output features exactly. Such a weight extraction technique enforces a strict notion of weight sharing between architectures, regardless of the global architecture information (e.g., different number of features for all the other layers) of these architectures. For instance, architectures $a$ and $b$ can have

---

[1]Here we assume the number of input features does not change. If it will change, then only the first several columns of $W_a$ are extracted.

widely different model capacities (e.g., $5M$ vs $90M$ number of architecture-specific parameters). The smaller architecture (e.g., $5M$) has to share all its weights with the other architecture (e.g., $90M$) and the supernet (as modeled by $f_a(x; W)$) cannot allocate any weights that are specific to the smaller architecture only. Another problem with $f_a(x; W)$ is that the overall capacity of the supernet is bounded by the number of parameters in the largest subnetwork (i.e. $W$) from the search space. However, the supernet weights $W$ need to parameterize a large amount of different subnetworks in the search space. *This is a fundamental limitation of the standard weight sharing mechanism. Section 3.1 proposes a reformulation to address this limitation, which is instantiated using two methods (Layer-wise MoS, Section 3.2, Neuron-wise MoS, Section 3.3) and can be dropped into Transformers (see Section 3.4).*

### 3.1 GENERALIZED MODEL FUNCTION

We can reformulate the function $f_a(x; W)$ to a generalized form $g(x, a; E)$, which takes 2 inputs: the input data $x$, and the activated architecture $a$. $E$ includes the learnable parameters of $g$. Then, the training objective of the proposed supernet becomes,

$$\min_E \mathbb{E}_{x,y \sim \mathcal{X}_{tr}} \mathbb{E}_{a_{rand} \sim \mathcal{A}} [\mathcal{L}(g(x, a_{rand}; E), y)]. \quad (3)$$

For the standard weight sharing mechanism mentioned above, $E = W$ and function $g$ just uses $a$ to perform the "trimming" operation on the weight matrix $W$, and forwards the subnetwork. To further minimize the objective equation 3, one feasible way is improving the capacity of the model function $g$. However, common ways such as adding hidden layers or hidden neurons are not applicable here, as we cannot change the final subnetwork architecture of mapping $x$ to $f_a(x; W)$. In this work, we propose to use the idea of Mixture-of-Experts (MoE) (Fedus et al., 2022) to improve the capacity of $g$. Specifically, we dynamically generate the weights $W_a$ according to specific architecture $a$ by routing to certain weights matrices from a set of expert weights. We call this architecture-routed MoE based supernet *Mixture-of-Supernets (MoS)*, and design two routing mechanisms for function $g(x, a; E)$. Due to lack of space, the detailed algorithm for supernet training and search is shown in A.2.

### 3.2 LAYER-WISE MOS

Assume there are $m$ (number of experts) unique weight matrices ($\{E^i \in \mathcal{R}^{n_{out_{big}} \times n_{in_{big}}}\}_{i=1}^m$, or expert weights), which are learnable parameters. For simplicity, we only use a single linear layer as the example. For an architecture $a$ with $n_{out_a}$ output features, we propose the layer-wise MoS that computes the weights specific to the architecture $a$ (i.e. $W_a \in \mathcal{R}^{n_{out_a} \times n_{in}}$) by performing a weighted combination of expert weights, $W_a = \sum_i \alpha_a^i E_a^i$. Here, $E_a^i \in \mathcal{R}^{n_{out_a} \times n_{in}}$ corresponds to the standard top rows extraction from the $i^{th}$ expert weights. The alignment vector ($\alpha_a \in [0, 1]^m, \sum_i \alpha_a^i = 1$) captures the alignment scores of the architecture $a$ with respect to each expert (weights matrix). We encode the architecture $a$ as a numeric vector $\text{Enc}(a) \in \mathcal{R}^{n_{enc} \times 1}$ (e.g., a list of the number of output features for different layers), and apply a learnable router $r(\cdot)$ (an MLP with softmax) to produce such scores, i.e. $\alpha_a = r(\text{Enc}(a))$. Thus, the generalized model function for the linear layer (as shown in Figure 1 (b)) can be defined as (omitting bias for brevity):

$$g(x, a; E) = W_a x = \sum_i r(\text{Enc}(a))^i E_a^i x. \quad (4)$$

Router $r(\cdot)$ controls the degree of weight sharing (unsharing) between two architectures by modulating the alignment scores ($\alpha_a$). For example, if $m = 2$ and $a$ is a subnetwork of the architecture $b$, the supernet could allocate weights that are specific to the smaller architecture $a$ only by setting $\alpha_a = (1, 0)$ and $\alpha_b = (0, 1)$. In this case, $g(x, a; E)$ only uses weights from $E_1$ and $g(x, b; E)$ only uses weights from $E_2$, so $E_1$ and $E_2$ can be updated towards the loss from architecture $a$ and $b$ without conflicts. It should be noted that few-shot NAS (Zhao et al., 2021c) can be seen as a special case of our framework if the router $r$ is rule-based. In addition, $g(\cdot)$ is essentially an MoE so that it has stronger expressive power and can lead the objective equation 3 to be smaller. After the supernet training completes, given an architecture $a$, the score $\alpha_a = r(\text{Enc}(a))$ can be generated offline. Expert weights are collapsed and the resulting number of parameters for the architecture $a$ becomes $n_{out_a} \times n_{in_a}$. Layer-wise MoS induces low degree of weight sharing between differently sized architectures shown by higher Jensen-Shannon distance between their alignment probability vectors compared to that of similarly sized architectures. See A.1.1 for more details.

| Supernet | MNLI | CoLA | MRPC | SST2 | QNLI | QQP | RTE | Avg. GLUE (↑) |
|---|---|---|---|---|---|---|---|---|
| Standalone | 82.61 | **59.03** | 86.54 | 91.52 | 89.47 | 90.68 | 71.53 | 81.63 |
| Supernet (Sandwich) | 82.34 | 57.58 | 86.54 | 91.74 | 88.67 | 90.39 | 73.26 | 81.50 (-0.13) |
| Layer-wise MoS (ours) | 82.40 | 57.62 | 87.26 | 92.08 | **89.57** | **90.68** | **77.08** | 82.38 (+0.75) |
| Neuron-wise MoS (ours) | **82.68** | 58.71 | 87.74 | **92.16** | 89.22 | 90.49 | 76.39 | **82.48 (+0.85)** |

Table 2: GLUE validation performance of different supernets (0 additional pretraining steps) compared to standalone (1x pretraining budget). The BERT architecture ($67M$ parameters) is the top model from the pareto front of Supernet (Sandwich) on SuperShaper's search space. Improvement (%) in GLUE average over standalone is enclosed in parentheses in the last column. Layer-wise and neuron-wise MoS perform significantly better than standalone.

### 3.3 NEURON-WISE MOS

The layer-wise MoS follows a conventional MoE setup, i.e., each expert is a linear layer/module. The router decides to use which experts combination to forward the input $x$ to, depending on $a$. In this case, the degree of freedom of weights generation is $m$, and the number of parameters grows by $m \times |W|$, where $|W|$ denotes the number of parameters in the standard supernet. Thus we need $m$ to be large enough to keep a good flexibility for the subnetwork weights generation, but this will also introduce too many parameters into the supernet and make the layer-wise MoS hard to train. This motivates us to use a smaller granularity of weights to represent each expert. Specifically, we use neurons in DNN as experts. In terms of the weight matrix, neuron-wise MoS uses one row of matrix to represent an individual expert. In contrast, layer-wise MoS uses an entire weight matrix. For neuron-wise MoS, the router output $\beta_a = r(\cdot) \in [0, 1]^{n_{out_{big}} \times m}$ for each layer, and the sum of each row in $\beta_a$ is 1. Similar to layer-wise MoS, we use an MLP to produce the $n_{out_{big}} \times m$ matrix and apply softmax on each row. We formulate the function $g(x, a; E)$ for neuron-wise MoS as

$$W_a = \sum_i \text{diag}(\beta_a^i) E_a^i, \tag{5}$$

where $\text{diag}(\beta)$ constructs a $n_{out_{big}} \times n_{out_{big}}$ diagonal matrix by putting $\beta$ on the diagonal, and $\beta_a^i$ is the $i$-th column of $\beta_a$. $E^i$ is still an $n_{out_{big}} \times n_{in}$ matrix as in layer-wise MoS. Compared to the layer-wise MoS, the neuron-wise MoS has more flexibility ($m \times n_{out_a}$ instead of only $m$) to control the degree of weight sharing between different architectures, while the number of parameters is still proportional to $m$. Neuron-wise MoS provides a more fine-grained control of weight sharing between subnetworks. We compute gradient conflict using cosine similarity between the supernet gradient and the smallest subnet gradient, following NASVIT work (Gong et al., 2021). As discussed in A.1.2, we find that Neuron-wise MoS enjoys lowest gradient conflict compared to Layer-wise MoS and HAT, shown by highest cosine similarity.

### 3.4 ADDING $g(x, a; E)$ TO TRANSFORMER

MoS is applicable to a single linear layer, multiple linear layers, and other parameterized layers (e.g., layer-norm). Since the linear layer dominates the number of parameters, we follow the approach used in most MoE work (Fedus et al., 2022). We take the standard weight-sharing based Transformer ($f_a(x; W)$) and replace the two linear layers in every feed-forward network block with $g(x, a; E)$.

## 4 EXPERIMENTS - EFFICIENT BERT

### 4.1 EXPERIMENT SETUP

We discuss application of our proposed supernet for building efficient task-agnostic BERT (Devlin et al., 2019) models. We focus on the BERT pretraining task, where a language model is pretrained from scratch to learn task-agnostic text representations using a masked language modeling objective. The pretrained BERT model can then be directly finetuned on several downstream NLP tasks. We focus on building BERT models that are highly accurate yet small (e.g., $5M - 50M$ parameters). BERT supernet and standalone are pretrained from scratch on Wikipedia and Books Corpus (Zhu et al., 2015). We evaluate the performance of the BERT model by finetuning on each of the

| Supernet | #Params | #Steps | CoLA | MRPC | SST2 | QNLI | QQP | RTE | Avg. GLUE |
|---|---|---|---|---|---|---|---|---|---|
| NAS-BERT | 5M | 125K | 19.8 | 79.6 | **87.3** | 84.9 | 85.8 | 66.7 | 70.7 |
| AutoDistil (proxy) | 6.88M | 0 | 24.8 | 78.5 | 85.9 | **86.4** | **89.1** | 64.3 | 71.5 |
| Neuron-wise MoS | 5M | 0 | **28.3** | **82.7** | 86.9 | 84.1 | 88.5 | **68.1** | **73.1** |
| NAS-BERT | 10M | 125K | 34.0 | 79.1 | **88.6** | 86.3 | 88.5 | **66.7** | 73.9 |
| Neuron-wise MoS | 10M | 0 | **34.7** | **81.0** | 88.1 | 85.1 | **89.1** | 66.7 | **74.1** |
| AutoDistil (proxy) | 26.1M | 0 | 48.3 | **88.3** | 90.1 | **90.0** | 90.6 | 69.4 | 79.5 |
| AutoDistil (agnostic) | 26.8M | 0 | 47.1 | 87.3 | **90.6** | 89.9 | **90.8** | 69.0 | 79.1 |
| Neuron-wise MoS | 26.8M | 0 | **52.7** | 88.0 | 90.0 | 87.7 | 89.9 | **78.1** | **81.1** |
| NAS-BERT | 30M | 125K | 48.7 | 84.6 | 90.5 | **88.4** | **90.2** | 71.8 | 79.0 |
| Neuron-wise MoS | 30M | 0 | **51.0** | **87.3** | **91.1** | 87.9 | **90.2** | **72.2** | **80.0** |
| AutoDistil (proxy) | 50.1M | 0 | **55.0** | **88.8** | 91.1 | **90.8** | **91.1** | 71.9 | 81.4 |
| Neuron-wise MoS | 50M | 0 | **55.0** | 88.0 | **91.9** | 89.0 | 90.6 | **75.4** | **81.6** |

Table 3: Comparison of neuron-wise MoS with NAS-BERT and AutoDistil for different model sizes ($\leq 50M$ parameters) based on GLUE validation performance. Neuron-wise MoS use a search space of $550$ architectures, which is on par with AutoDistil. The third column corresponds to the number of additional training steps required to obtain the weights for the final architecture after supernet training. Performance numbers for the baseline models are taken from the corresponding papers. On average GLUE, neuron-wise MoS can perform similarly or improves over NAS-BERT for different model sizes without any additional training. Neuron-wise MoS can improve over AutoDistil for most model sizes in average GLUE. See A.3.3 for the hyperparameters of best architectures.

seven tasks (chosen by AutoDistil (Xu et al., 2022a)) in the GLUE benchmark (Wang et al., 2018). The architecture encoding, data preprocessing, pretraining settings, and finetuning settings are discussed in A.3.1. The baseline models are standalone and standard supernet as proposed in SuperShaper (Ganesan et al., 2021). Our proposed models are layer-wise and neuron-wise MoS. All the supernets are trained using sandwich training. [2] The parameters $m$ and router's hidden dimension are set to 2 and 128, respectively, for MoS supernets.

## 4.2 SUPERNET VS. STANDALONE GAP

For studying the supernet vs. the standalone gap, the search space is taken from SuperShaper (Ganesan et al., 2021), which consists of BERT architectures that vary only in the hidden size at each layer (*{120, 240, 360, 480, 540, 600, 768}*) with fixed number of layers (12) and attention heads (12). The search space amounts to around $14B$ architectures. We study the supernet vs. the standalone model gap for the top model architecture from the pareto front of Supernet (Sandwich) (Ganesan et al., 2021). Table 2 displays the GLUE benchmark performance of standalone training of the architecture (1x pretraining budget, which is 2048 batch size * 125,000 steps) as well as architecture-specific weights from different supernets (0 additional pretraining steps; that is, only supernet pretraining). The gap between the task-specific supernet and the standalone performance is bridged by MoS (layer-wise or neuron-wise) for 6 out of 7 tasks, including MNLI (which is a widely used task to indicate performance of a pretrained language model (Liu et al., 2019; Xu et al., 2022b)). The gap in average GLUE between the standalone model and the standard supernet is $0.13$ points. Notably, equipped with customization and expressivity properties, the layer-wise and neuron-wise MoS significantly improve upon the standalone training by $0.75$ and $0.85$ average GLUE points, respectively.

## 4.3 COMPARISON WITH SOTA NAS

The state-of-the-art NAS frameworks for building a task-agnostic BERT model are NAS-BERT (Xu et al., 2021) and AutoDistil (Xu et al., 2022a). [3] The NAS-BERT pipeline includes: (1) supernet training (with a Transformer stack containing multi-head attention, feed-forward network [FFN] and convolutional layers in arbitrary positions), (2) search based on the distillation (task-agnostic) loss, and (3) pretraining the best architecture from scratch (1x pretraining budget, which is 2048 batch

---

[2]SuperShaper (Ganesan et al., 2021) observe that SPOS performs poorly compared to sandwich training. Hence, we do not study SPOS for building BERT models. The learning curve is shown in A.3.2.

[3]AutoDistil (proxy) outperforms SOTA distillation approaches such as TinyBERT (Jiao et al., 2020) and MINILM (Wang et al., 2020b) by $0.7$ average GLUE points. Hence, we do not compare against these works.

size * 125,000 steps). The third step has to be executed for every constraint change and hardware change, which is very expensive. AutoDistil pipeline includes: (1) construct $K$ search spaces and train supernets for each search space independently, (2a) *agnostic*-search mode: search based on the self-attention distillation (task-agnostic) loss, (2b) *proxy*-search mode: search based on the MNLI validation score, and (3) extract the architecture-specific weights from the supernet without additional training. The first step can be expensive as pretraining $K$ supernets can take $K$ times training compute and memory, compared to training a single supernet. The proxy-search model can unfairly benefit AutoDistil, as it finetunes all the architectures in its search space on MNLI and uses the MNLI score to rank the architectures. For fair comparison with SOTA, we exclude MNLI task from evaluation. [4]

Our proposed NAS pipeline overcomes all the issues with NAS-BERT and AutoDistil. For comparison with the SOTA NAS, our search space contains BERT architectures with homogeneous Transformer layers: hidden size (120 to 768 in increments of 12), attention heads ({*6, 12*}), intermediate FFN hidden dimension ratio ({*2, 2.5, 3, 3.5, 4*}). This search space amounts to 550 architectures, which is on par with AutoDistil. The supernet is based on neuron-wise MoS. The search uses the perplexity (task-agnostic) metric to rank the architectures. Unlike NAS-BERT which pretrains the best architecture from scratch (third step), the final architecture weights are directly extracted from the supernet without further pretraining. Unlike AutoDistil which pretrains $K$ supernets, the proposed pipeline pretrains exactly one supernet, which requires significantly less training compute and memory. Unlike AutoDistil's proxy setting where MNLI performance guides the search, our proposed pipeline uses only task-agnostic metric (like AutoDistil's agnostic). Table 3 shows the comparison of neuron-wise MoS based supernet with NAS-BERT and AutoDistil for different model sizes. The performance of NAS-BERT and AutoDistil are taken from the corresponding papers. On average GLUE, our proposed pipeline: (i) improves over NAS-BERT for $5M$, $10M$, and $30M$ model sizes, without any additional training (100% additional training compute savings, which is 2048 batch size * 125,000 steps). On average GLUE, our proposed pipeline: (i) improves over AutoDistil-proxy for $6.88M$ and $50M$ model sizes respectively with $1.88M$ and $0.1M$ fewer parameters respectively and (ii) improves over both AutoDistil-proxy and AutoDistil-agnostic for $26M$ model size. Besides achieving SOTA results, the main benefit of our method is reducing the heavy workload of training multiple models in either subnetwork retraining (NAS-BERT) or supernet training (AutoDistil).

## 5 EXPERIMENTS - EFFICIENT MACHINE TRANSLATION

### 5.1 EXPERIMENT SETUP

In this section, we discuss the application of proposed supernets for building efficient MT models. We follow the experimental setup provided by Hardware-aware Transformers (HAT (Wang et al., 2020a)), which is the SOTA NAS framework for building MT models that enjoy good latency-BLEU tradeoffs. We focus on three popular MT benchmarks (Bojar et al., 2014; Wikimedia-Foundation, 2019): WMT'14 En-De, WMT'14 En-Fr and WMT'19 En-De, whose dataset statistics are shown in A.4.1. The architecture encoding, training settings for both supernet and standalone models are the same, which are discussed in A.4.2. The baseline supernets include: (i) HAT – HAT's supernet that uses single path one-shot optimization, and (ii) Supernet (Sandwich) – Supernet that uses sandwich training. The proposed supernets include: (i) Layer-wise MoS – MoS with layer-wise routing and sandwich training and (ii) Neuron-wise MoS – MoS with neuron-wise routing and sandwich training. The parameters $m$ and router's hidden dimension are set to 2 and 128 respectively for both MoS variants. See A.4.8 for the rationale behind the choice of '$m$'.

### 5.2 SUPERNET VS. STANDALONE GAP

HAT's search space consists of $6M$ encoder-decoder architectures, with flexible embedding size (512 or 640), decoder layers (1 to 6), self / cross attention heads (4 or 8), and number of top encoder layers for the decoder to attend to (1 to 3). For a given architecture, supernet performance corresponds to evaluating the architecture-specific weights extracted from the supernet, while standalone performance corresponds to evaluating the architecture after training from scratch. For a random sample of

---

[4]See A.3.4 for the comparison of neuron-wise MoS against baselines that do not directly tune on the MNLI task, where we find that neuron-wise MoS improves over baselines consistently in terms of both average GLUE and MNLI task performance.

| Dataset | WMT'14 En-De | | WMT'14 En-Fr | | WMT'19 En-De | |
|---|---|---|---|---|---|---|
| Supernet | MAE ($\downarrow$) | Kendall ($\uparrow$) | MAE ($\downarrow$) | Kendall ($\uparrow$) | MAE ($\downarrow$) | Kendall ($\uparrow$) |
| HAT | 1.84 | 0.81 | 1.37 | 0.63 | 2.07 | 0.71 |
| Supernet (Sandwich) | 1.62 (12%) | 0.81 | 1.37 (0%) | 0.63 | 2.02 (2.4%) | 0.87 |
| Layer-wise MoS (ours) | 1.61 (12.5%) | 0.54 | 1.24 (9.5%) | 0.73 | 1.57 (24.2%) | 0.87 |
| Neuron-wise MoS (ours) | **1.13 (38.6%)** | 0.71 | **1.2 (12.4%)** | 0.85 | **1.48 (28.5%)** | 0.81 |

Table 4: Mean absolute error (MAE) and Kendall rank correlation coefficient between the supernet and the standalone model BLEU performance for 15 random architectures from the MT search space. Improvements (%) in mean absolute error over HAT are in parentheses. Our supernets enjoy minimal MAE and comparable ranking quality with respect to the baseline models.

| Dataset | WMT'14 En-De | | | WMT'14 En-Fr | | | WMT'19 En-De | | |
|---|---|---|---|---|---|---|---|---|---|
| Supernet / Latency Constraint | 100 ms | 150 ms | 200 ms | 100 ms | 150 ms | 200 ms | 100 ms | 150 ms | 200 ms |
| HAT | 25.26 | 26.25 | 26.28 | 38.94 | 39.26 | 39.16 | 42.61 | 43.07 | 43.23 |
| Layer-wise MoS (ours) | 26.28 | 27.31 | **28.03** | 39.34 | **40.29** | **41.24** | 43.45 | **44.71** | 46.18 |
| Neuron-wise MoS (ours) | **26.37** | **27.59** | 27.79 | **39.55** | 40.02 | 41.04 | **43.77** | 44.66 | **46.21** |

Table 5: Latency vs. Supernet BLEU for the models on the pareto front, obtained by performing search with different latency constraints (100 ms, 150 ms, 200 ms) on the NVIDIA V100 GPU. Our supernets yield architectures that enjoy better latency-BLEU tradeoffs than HAT.

architectures from the search space, a good supernet must have: (i) minimal mean absolute error (MAE) and (ii) high rank correlation between the standalone and the supernet performance. Table 4 shows the mean absolute error and Kendall rank correlation coefficient for 15 random architectures from the search space. Compared to HAT, supernet with sandwich training has better MAE and rank quality. This result highlights that sandwich training is essential for building good supernet compared to SPOS for machine translation. Compared to the supernet with sandwich training, our proposed supernets achieve comparable ranking quality for WMT'14 En-Fr and WMT'19 En-De tasks, while marginally underperforming for WMT'14 En-De task. Our proposed supernets achieve minimal MAE on all the three tasks. Specifically, neuron-wise MoS obtains the biggest MAE improvements, which suggests that additional training steps required to make MAE negligible might be the lowest for neuron-wise MoS among all the supernet variants (as we show in Section 5.4). We also plot the supernet and the standalone performance for each architecture, where we find that neuron-wise MoS particularly excels for almost all the top performing architectures (see A.4.3). The training overhead for MoS is generally negligible. For example, for WMT'14 En-De task, the supernet training time (single NVIDIA V100) is 248 hours, while neuron-wise MoS and layer-wise MoS require additional hours of 14 and 18 hours respectively (less than 8% overhead, see Section A.4.10 for details).

## 5.3 COMPARISON WITH SOTA NAS

The pareto front from the supernet can be obtained using the evolutionary search algorithm, which takes the supernet for quickly identifying the top performing candidate architectures, and the latency estimator, which can quickly discard candidate architectures that have latencies exceeding latency threshold. The settings for the evolutionary search algorithm and the latency estimator can be seen in A.4.4. We experiment with 3 latency thresholds: 100 ms, 150 ms, and 200 ms. Table 5 shows the latency vs. the supernet performance tradeoff for the models in the pareto front from different supernets. Compared to HAT, the proposed supernets achieve significantly higher BLEU for each latency threshold across all the datasets, which highlights the importance of architecture specialization and expressiveness of the supernet. See A.4.6 for the consistency of these trends for different seeds.

## 5.4 ADDITIONAL TRAINING TO CLOSE THE GAP

The proposed supernets minimize the supernet vs. the standalone MAE gap significantly (as discussed in Section 5.2), but still do not make the gap negligible. To close the gap for an architecture, one need to extract the architecture-specific weights from the supernet and perform additional training until the standalone performance is reached (when the gap becomes 0). A good supernet should require minimal number of additional steps and time for the architectures extracted from the supernet to close

| Dataset Supernet | Additional training steps (↓) | | | Additional training time (NVIDIA V100 hours) (↓) | | |
|---|---|---|---|---|---|---|
| | WMT'14 En-De | WMT'14 En-Fr | WMT'19 En-De | WMT'14 En-De | WMT'14 En-Fr | WMT'19 En-De |
| HAT | 33K | 33K | 26K | 63.9 | **60.1** | 52.3 |
| Laye. MoS | 16K (51.5%) | 30K (9%) | 20K (23%) | 35.5 (44.4%) | 66.5 (-10.6%) | 45.2 (13.5%) |
| Neur. MoS | **13K (60%)** | **26K (21%)** | **16K (38.4%)** | **31.0 (51.4%)** | 61.7 (-2.7%) | **39.5 (24.5%)** |

Table 6: Average number of additional training steps and time required for the models on the pareto front to close the supernet vs. standalone gap. Improvements (%) over HAT are shown in parentheses. Our supernets require minimal number of additional training steps and time to close the gap compared to HAT for most tasks. See A.4.5 for each latency constraint.

the gap. For additional training, we evaluate the test BLEU of each architecture after every $10K$ steps and stop when the test BLEU matches or exceeds the test BLEU of the standalone model. Table 6 displays the average number of additional training required for all the models on the pareto front from each supernet to close the gap. Compared to HAT, layer-wise MoS provides an impressive reduction of 9% to 51% in training steps, while neuron-wise MoS provides by far the largest reduction of 21% to 60%. For the WMT'14 En-Fr task, both MoS supernets require at least 2.7% more time than HAT to achieve SOTA BLEU across different constraints. These results highlight that architecture specialization and supernet expressivity are crucial in greatly improving training efficiency of the subnets extracted from the supernet.

## 6 RELATED WORK

In this section we briefly discuss existing NAS research in NLP. Evolved Transformer (ET) (So et al., 2019) is an initial work that searches for efficient MT models using NAS. It uses evolutionary search which can dynamically allocate training resources for promising candidates. ET requires $2M$ GPU hours. HAT (Wang et al., 2020a) propose a weight-sharing supernet as performance estimator. HAT uses supernet to amortize training cost for candidate MT evaluations needed by evolutionary search, which reduces overall search cost by 12000x compared to ET. NAS-BERT (Xu et al., 2021) partitions the BERT-Base model into blocks and trains a weight-sharing supernet to distill each block. During supernet training, NAS-BERT prunes less promising candidates from the search space using progressive shrinking. It can quickly identify the top architecture for each efficiency constraint. NAS-BERT needs to pretrain the top architecture from scratch for every constraint change, which can be very expensive. SuperShaper (Ganesan et al., 2021) pretrains a weight-sharing supernet for BERT using masked language modeling objective with sandwich training. The authors find that SPOS performs poorly compared to the sandwich training objective. AutoDistil (Xu et al., 2022a) employs few-shot NAS (Zhao et al., 2021b): construct $K$ search spaces of non-overlapping BERT architectures and train a weight-sharing BERT supernet for each search space. The search is based on self-attention distillation loss with BERT-Base (task-agnostic search) and MNLI score (proxy search).

In computer vision community, K-shot NAS (Su et al., 2021) generates the weight for each subnet as a convex combination of different supernet weights in a dictionary with a simplex code. Their framework is similar to layer-wise MoS with the following key differences. K-shot NAS trains the architecture code generator and supernet iteratively due to training difficulty, while layer-wise MoS trains all its components jointly. K-shot NAS has been applied only in convolutional architectures for image classification tasks. K-shot NAS introduces too many parameters with increase in number of supernets ($K$), which is alleviated by neuron-wise MoS due to its granular weight specialization. In this work, we focus on tasks in NLP (and the relevant baselines), where we find that the supernets lag behind standalone models significantly in terms of performance. Also, authors of k-shot NAS do not release the code to reproduce their results. Hence, we do not evaluate against k-shot NAS.

## 7 CONCLUSION

In this work, we proposed Mixture-of-Supernets, a formulation to improve supernet by enhancing its expressive power. We showed that the idea of MoE can be adopted to generate flexible weights for subnetworks. From our extensive evaluation for building efficient BERT and MT models, we showed that our supernets can: (i) minimize the retraining time thereby improving the NAS efficiency significantly and (ii) yield high quality architectures satisfying user-defined constraints via NAS.

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

| Model 1 | Model 2 | WMT'14 En-De | WMT'14 En-Fr | WMT'19 En-De |
|---|---|---|---|---|
| Smallest A ($23M$) | Largest A ($118M$) | 0.297 | 0.275 | 0.263 |
| Smallest B ($23M$) | Largest B ($118M$) | 0.281 | 0.258 | 0.245 |
| Smallest A ($23M$) | Largest B ($118M$) | 0.284 | 0.263 | 0.249 |
| Smallest B ($23M$) | Largest A ($118M$) | 0.294 | 0.27 | 0.259 |
| Smallest A ($23M$) | Smallest B ($118M$) | 0.006 | 0.008 | 0.004 |
| Largest A ($23M$) | Largest B ($118M$) | 0.014 | 0.012 | 0.015 |

Table 7: Jensen-Shannon distance of Layer-wise MoS alignment vector across models as a weight sharing measure. Layer-wise MoS induces low degree of weight sharing between differently sized architectures shown by higher Jensen-Shannon distance between their alignment vectors compared to that of similarly sized architectures. Note that architectures A and B differ by number of encoder/decoder attention heads.

| Supernet | WMT'14 En-De | WMT'19 En-De |
|---|---|---|
| HAT | 0.522 | 0.416 |
| Layer-wise MoS | 0.515 | 0.517 |
| Neuron-wise MoS | 0.555 | 0.52 |

Table 8: Gradient conflict via cosine similarity between the supernet gradient and the smallest subnet gradient. Neuron-wise MoS enjoys lower gradient conflict, shown via. high cosine similarity.

# A APPENDIX

## A.1 WEIGHT SHARING AND GRADIENT CONFLICT ANALYSIS

### A.1.1 JENSEN-SHANNON DISTANCE OF ALIGNMENT VECTOR AS A WEIGHT SHARING MEASURE

We use the Jensen-Shannon distance of alignment vector generated by Layer-wise MoS for two architectures as a proxy to quantify the degree of weight sharing. Ideally, the lower the Jensen-Shannon distance, the higher the degree of weight sharing and vice-versa. We experiment with two architectures of 23M parameters (Smallest A and Smallest B) and two architectures of 118M parameters (Largest A and Largest B). From Table 7, it is clear that Layer-wise MoS induces low degree of weight sharing between differently sized architectures shown by higher Jensen-Shannon distance between their alignment vectors. On the other hand, there is a high degree of weight sharing between similarly sized architectures where Jensen-Shannon distance is significantly low.

### A.1.2 COSINE SIMILARITY BETWEEN THE SUPERNET GRADIENT AND THE SMALLEST SUBNET GRADIENT AS A GRADIENT CONFLICT MEASURE.

We compute gradient conflict using cosine similarity between the supernet gradient and the smallest subnet gradient, following NASVIT work (Gong et al., 2021). In Table 8, we show that Neuron-wise MoS enjoys lowest gradient conflict compared to Layer-wise MoS and HAT, shown by highest cosine similarity.

## A.2 DETAILED ALGORITHM FOR SUPERNET TRAINING AND SEARCH

### A.2.1 SUPERNET TRAINING ALGORITHM

The detailed algorithm for supernet training is shown in Algorithm 1.

### A.2.2 SEARCH ALGORITHM

The detailed algorithm for search is shown in Algorithm 2.

**Algorithm 1** Training algorithm for Mixture-of-Supernets used in MT.

**Input:** `Training data:` $\mathcal{X}_{tr}$, `Search space:` $\mathcal{A}$,
`No. of training steps: num-train-steps,` `Type of MoS: mos-type`
**Output:** `Training Supernet Weights:` $\mathbb{E}$

1: $\mathbb{E} \leftarrow$ Random weights from Normal Distribution.
2: **for** $iter \leftarrow 1$ `to num-train-steps` **do**
3:      // sample data
4:      $x, y \sim \mathcal{X}_{tr}$
5:      // perform sandwich sampling
6:      **for** a in $[a_{rand} \sim \mathcal{A}, a_{big}, a_{small}]$ **do**
7:          Enc(a) // create the architecture encoding
8:          // generate architecture-specific weights
9:          **if** mos-type == Layer wise MoS **then**
10:              $W_a = \sum_i r(\text{Enc}(a))^i E_a^i$
11:          **else if** mos-type == Neuron wise MoS **then**
12:              $W_a = \sum_i \text{diag}(\beta_a^i) E_a^i$
13:      // compute task-specific loss
14:      loss $\leftarrow \mathcal{L}(W_a x, y)$
15:      loss.backward() // compute gradients
16:      Update $\mathbb{E}$ using accumulated gradients // learning rule
17: return $\mathbb{E}$

**Algorithm 2** Evolutionary search algorithm for Neural architecture search used in MT.

**Input:** `supernet, latency-predictor, num-iterations, num-population,`
`num-parents, num-mutations, num-crossover, mutate-prob,`
`latency-constraint`
**Output:** `best-architecture`

1: // create initial population
2: $popu \leftarrow$ `num-population` random samples from the search space
3: **for** $iter \leftarrow 1$ `to num-iterations` **do**
4:      // generate parents by picking top candidates
5:      `cur-parents` $\leftarrow$ top 'num-parents' architectures from $popu$ by MoS validation loss
6:      // generate candidates via mutation
7:      `cur-mutate-popu` $= \{\}$
8:      **for** $mi \leftarrow 1$ `to num-mutations` **do**
9:          `cur-mutate-gene` $\leftarrow$ mutate a random example from $popu$ with mutation probability `mutate-prob`
10:          **if** `cur-mutate-gene` satisfies `latency-constraint` via `latency-predictor` **then**
11:              `cur-mutate-popu = cur-mutate-popu` $\cup$ `cur-mutate-gene`
12:      // generate candidates via cross-over
13:      `cur-crossover-popu` $= \{\}$
14:      **for** $ci \leftarrow 1$ `to num-crossover` **do**
15:          `cur-crossover-gene` $\leftarrow$ crossover two random examples from $popu$
16:          **if** `cur-crossover-gene` satisfies `latency-constraint` via `latency-predictor` **then**
17:              `cur-crossover-popu` $=$ `cur-crossover-popu` $\cup$ `cur-crossover-gene`
18:      // update population
19:      $popu =$ `cur-parents` $\cup$ `cur-mutate-popu` $\cup$ `cur-crossover-popu`
20: return top architecture from $popu$ by MoS's validation loss

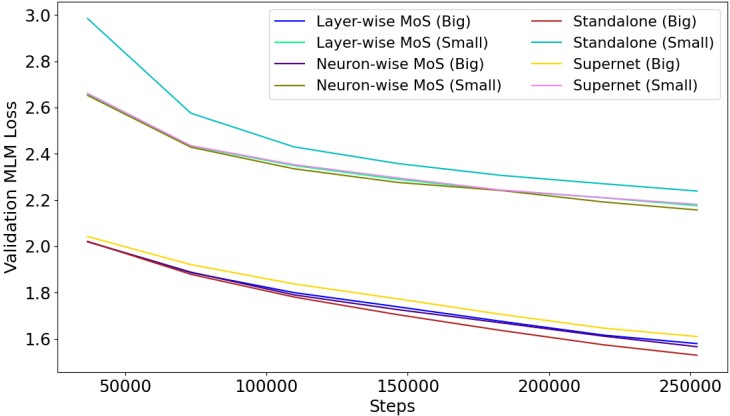

Figure 2: Learning Curve - Training steps vs. Validation MLM loss. 'Big' and 'Small' correspond to the largest and the smallest BERT architecture respectively from the search space of SuperShaper. 'Standalone' and 'Supernet' correspond to training from scratch and sampling from the supernet respectively. All the supernets are trained with sandwich training.

## A.3 ADDITIONAL EXPERIMENTS - EFFICIENT BERT

### A.3.1 BERT PRETRAINING / FINETUNING SETTINGS

**Pretraining data**: The pretraining data consists of text from Wikipedia and Books Corpus (Zhu et al., 2015). We use the data preprocessing scripts provided by Izsak et al. to construct the tokenized text.

**Supernet and standalone pretraining settings**: The pretraining settings for supernet and standalone models are taken from SuperShaper (Ganesan et al., 2021): batch size of 2048, maximum sequence length of 128, training steps of 125K, learning rate of $5e^{-4}$, weight decay of $0.01$, and warmup steps of $10K$ (0 for standalone). For experiments with the search space from Super-Shaper (Ganesan et al., 2021) (Section 4.2), the architecture encoding $a$ is a list of hidden size at each layer of the architecture (12 elements since the supernet is a 12 layer model). For experiments with the search space on par with AutoDistil (Xu et al., 2022a) (Section 4.3), the architecture encoding $a$ is a list of four elastic hyperparameters of the homogeneous BERT architecture: number of layers, hidden size of all layers, feedforward network (FFN) expansion ratio of all layers and number of attention heads of all layers (see Table 9 for sample homogeneous BERT architectures).

**Finetuning settings**: We evaluate the performance of the BERT model by finetuning on each of the seven tasks (chosen by AutoDistil (Xu et al., 2022a)) in the GLUE benchmark (Wang et al., 2018). The evaluation metric is the average accuracy (Matthews's correlation coefficient for CoLA only) on all the tasks (GLUE average). The finetuning settings are taken from the BERT paper (Devlin et al., 2019): learning rate from $\{5e^{-5}, 3e^{-5}, 2e^{-5}\}$, batch size from $\{16, 32\}$, and epochs from $\{2, 3, 4\}$.

### A.3.2 LEARNING CURVE FOR BERT SUPERNET VARIANTS

Figure 2 shows the training steps versus validation MLM loss (learning curve) for the standalone BERT model and different supernet based BERT variants. The standalone model and the supernet are compared for the biggest architecture (big) and the smallest architecture (small) from the search space of SuperShaper (Ganesan et al., 2021). For the biggest architecture, the standalone model performs the best. For the smallest architecture, the standalone model is outperformed by all the supernet variants. In both cases, the proposed supernets (especially neuron-wise MoS) perform much better than the standard supernet.

| Standalone / Supernet | Model Size | #Layers | #Hidden Size | #FFN Expansion Ratio | #Heads |
|---|---|---|---|---|---|
| BERT | 109M | 12 | 768 | 4 | 12 |
| AutoDistil (proxy) | 6.88M | 7 | 160 | 3.5 | 10 |
| Neuron-wise MoS | 5M | 12 | 120 | 2.0 | 6 |
| Neuron-wise MoS | 10M | 12 | 180 | 3.5 | 6 |
| AutoDistil (agnostic) | 26.8M | 11 | 352 | 4 | 10 |
| Neuron-wise MoS | 26.8M | 12 | 372 | 2.5 | 6 |
| Neuron-wise MoS | 30M | 12 | 384 | 3 | 6 |
| AutoDistil (proxy) | 50.1M | 12 | 544 | 3 | 9 |
| Neuron-wise MoS | 50M | 12 | 504 | 3.5 | 12 |

Table 9: Architecture comparison of the best architecture designed by the neuron-wise MoS with AutoDistil (Xu et al., 2022a) and BERT-Base (Devlin et al., 2019).

| Supernet | #Params | #Steps | MNLI | CoLA | MRPC | SST2 | QNLI | QQP | RTE | Avg. GLUE |
|---|---|---|---|---|---|---|---|---|---|---|
| NAS-BERT | 5M | 125K | 74.4 | 19.8 | 79.6 | **87.3** | **84.9** | 85.8 | 66.7 | 71.2 |
| Neuron-wise MoS | 5M | 0 | **75.5** | **28.3** | 82.7 | 86.9 | 84.1 | **88.5** | 68.1 | **73.4** |
| NAS-BERT | 10M | 125K | 76.4 | 34.0 | 79.1 | **88.6** | **86.3** | 88.5 | 66.7 | 74.2 |
| Neuron-wise MoS | 10M | 0 | **77.2** | **34.7** | 81.0 | 88.1 | 85.1 | **89.1** | 66.7 | **74.6** |
| AutoDistil (agnostic) | 26.8M | 0 | **82.8** | 47.1 | 87.3 | **90.6** | **89.9** | **90.8** | 69.0 | 79.6 |
| Neuron-wise MoS | 26.8M | 0 | 80.7 | **52.7** | **88.0** | 90.0 | 87.7 | 89.9 | **78.1** | **81.0** |
| NAS-BERT | 30M | 125K | 81.0 | 48.7 | 84.6 | 90.5 | **88.4** | 90.2 | 71.8 | 79.3 |
| Neuron-wise MoS | 30M | 0 | **81.6** | **51.0** | **87.3** | **91.1** | 87.9 | **90.2** | **72.2** | **80.2** |
| Neuron-wise MoS | 50M | 0 | 82.4 | 55.0 | 88.0 | 91.9 | 89.0 | 90.6 | 75.4 | 81.8 |

Table 10: Comparison of neuron-wise MoS with NAS-BERT and AutoDistil (agnostic) for different model sizes ($\leq 50M$ parameters) based on GLUE validation performance. We include results on MNLI task. For fair comparison, we drop AutoDistil (proxy), which directly uses MNLI task for architecture selection. Neuron-wise MoS improves over the baselines in all model sizes, in terms of average GLUE. For MNLI task, neuron-wise MoS improves over the baselines in most model sizes.

### A.3.3 ARCHITECTURE COMPARISON OF NEURON-WISE MOS VS. AUTODISTIL

Table 9 shows the comparison of the BERT architecture designed by our proposed neuron-wise MoS with AutoDistil.

### A.3.4 FAIR COMPARISON OF NEURON-WISE MOS W.R.T SOTA WITH MNLI

We compare neuron-wise MoS with NAS-BERT and AutoDistil (agnostic) for different model sizes ($\leq 50M$ parameters) based on GLUE validation performance. In Table 10, we include results on MNLI task. For fair comparison, we drop AutoDistil (proxy), which directly uses MNLI task for architecture selection. Neuron-wise MoS improves over the baselines in all model sizes, in terms of average GLUE. For MNLI task, neuron-wise MoS improves over the baselines in most model sizes.

## A.4 ADDITIONAL EXPERIMENTS - EFFICIENT MACHINE TRANSLATION

### A.4.1 MACHINE TRANSLATION BENCHMARK DATA

Table 11 shows the statistics of three machine translation datasets: WMT'14 En-De, WMT'14 En-Fr, and WMT'19 En-De.

### A.4.2 TRAINING SETTINGS AND METRICS

The training settings for both supernet and standalone models are the same: $40K$ training steps, Adam optimizer, a cosine learning rate scheduler, and a warmup of learning rate from $10^{-7}$ to $10^{-3}$ with cosine annealing. The best checkpoint is selected based on the validation loss, while the performance

| Dataset | Year | Source Lang | Target Lang | #Train | #Valid | #Test |
|---------|------|-------------|-------------|--------|--------|-------|
| WMT | 2014 | English (en) | German (de) | 4.5M | 3000 | 3000 |
| WMT | 2014 | English (en) | French (fr) | 35M | 26000 | 26000 |
| WMT | 2019 | English (en) | German (de) | 43M | 2900 | 2900 |

Table 11: Machine translation benchmark data.

of the MT model is evaluated based on BLEU. The beam size is four with length penalty of 0.6. The architecture encoding $a$ is a list of following 10 values:

1. *Encoder embedding dimension* corresponds to embedding dimension of the encoder.

2. *Encoder #layers* corresponds to number of encoder layers.

3. *Average encoder FFN. intermediate dimension* corresponds to average of FFN intermediate dimension across encoder layers.

4. *Average encoder self attention heads* corresponds to average of number of self attention heads across encoder layers.

5. *Decoder embedding dimension* corresponds to embedding dimension of the decoder.

6. *Decoder #Layers* corresponds to number of decoder layers.

7. *Average Decoder FFN. Intermediate Dimension* corresponds to average of FFN intermediate dimension across decoder layers.

8. *Average decoder self attention heads* corresponds to average of number of self attention heads across decoder layers.

9. *Average decoder cross attention heads* corresponds to average of number of cross attention heads across decoder layers.

10. *Average arbitrary encoder decoder attention* corresponds to average number of encoder layers attended by cross-attention heads in each decoder layer (-1 means only attend to the last layer, 1 means attend to the last two layers, 2 means attend to the last three layers).

### A.4.3 SUPERNET VS. STANDALONE PERFORMANCE PLOT

Figure 3 displays the supernet vs. the standalone performance for 15 randomly sampled architectures on all the three tasks. Neuron-wise MoS excel for almost all the top performing architectures ($\geq 26.5$ and $\geq 42.5$ standalone BLEU for WMT'14 En-De and WMT'19 En-De respectively), which indicates that the models especially in the pareto front can benefit immensely from neuron level specialization.

### A.4.4 HAT SETTINGS

**Evolutionary search**: The settings for the evolutionary search algorithm include: 30 iterations, population size of 125, parents population of 25, crossover population of 50, and mutation population of 50 with 0.3 mutation probability.

**Latency estimator**: The latency estimator is developed in two stages. First, the latency dataset is constructed by measuring the latency of 2000 randomly sampled architectures directly on the user-defined hardware (NVIDIA V100 GPU). Latency is the time taken to translate a source sentence to a target sentence (source and target sentence lengths of 30 tokens each). For each architecture, 300 latency measurements are taken, outliers (top 10% and bottom 10%) are removed, and the rest (80%) is averaged. Second, the latency estimator is a 3 layer multi-layer neural network based regressor, which is trained using encoding and latency of the architecture as features and labels respectively.

### A.4.5 ADDITIONAL TRAINING STEPS TO CLOSE THE GAP VS. PERFORMANCE

Figure 4, Figure 5, and Figure 6 show the additional training steps vs. BLEU for different latency constraints on the WMT'14 En-De task, WMT'14 En-Fr and WMT'19 En-De tasks respectively.

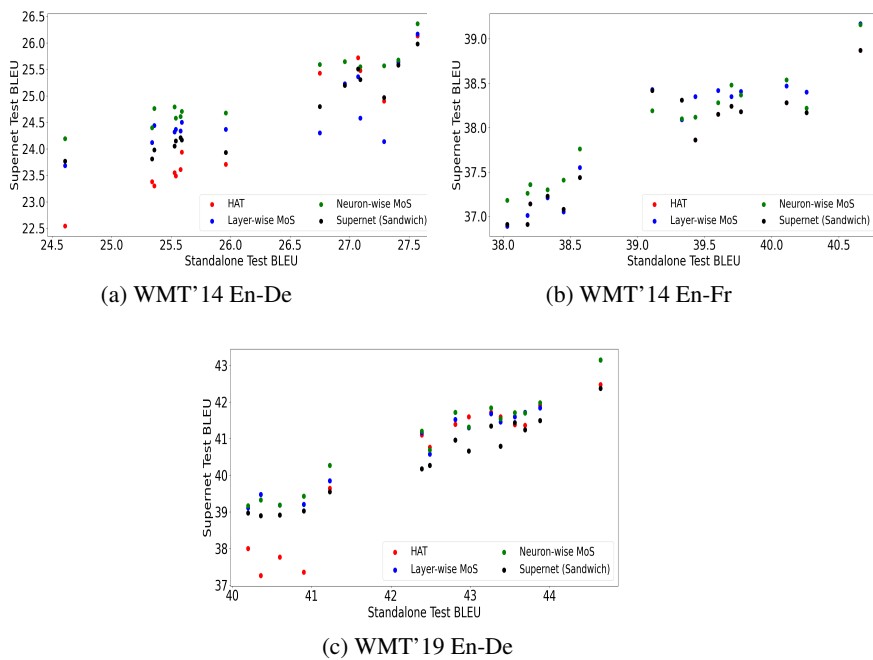

(a) WMT'14 En-De      (b) WMT'14 En-Fr

(c) WMT'19 En-De

Figure 3: Supernet vs. Standalone model performance for 15 random architectures from MT search space. Supernet performance is obtained by evaluating the architecture-specific weights extracted from the supernet. Standalone model performance is obtained by training the architecture from scratch to convergence and evaluating it.

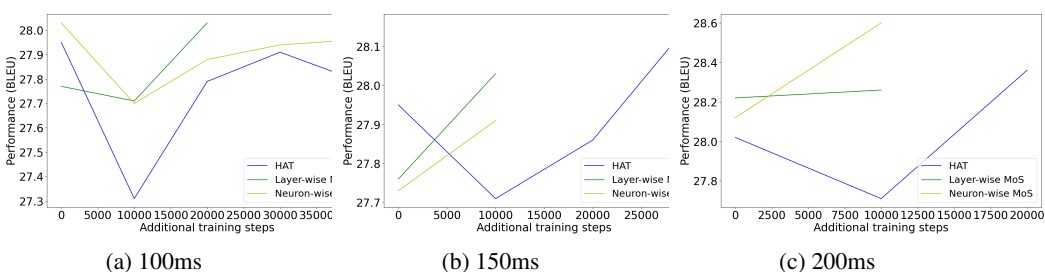

(a) 100ms      (b) 150ms      (c) 200ms

Figure 4: Additional training steps to close the supernet - standalone gap vs. performance for different latency constraints on the WMT'14 En-De dataset.

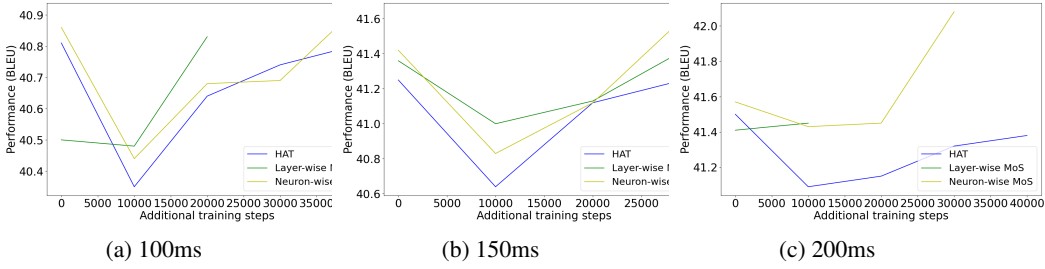

(a) 100ms      (b) 150ms      (c) 200ms

Figure 5: Additional training steps to close the supernet - standalone gap vs. performance for different latency constraints on the WMT'14 En-Fr dataset.

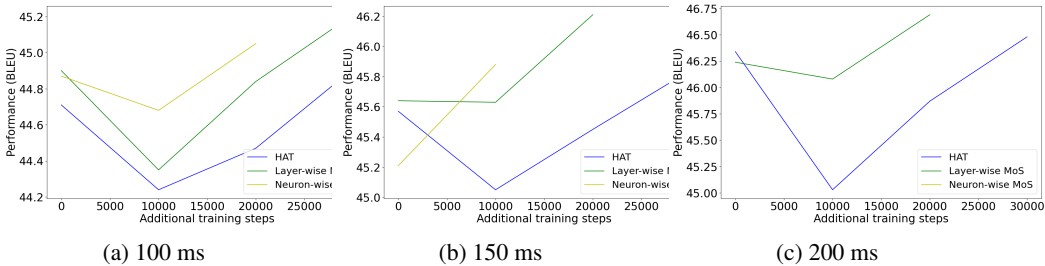

| (a) 100 ms | (b) 150 ms | (c) 200 ms |

Figure 6: Additional training steps to close the supernet - the standalone gap vs. performance for different latency constraints on the WMT'19 En-De dataset. For 200 ms latency constraint, neuron-wise MoS closes the gap without additional training.

| Supernet / Pareto Front | Seed | Model 1 | | Model 2 | | Model 3 | |
|---|---|---|---|---|---|---|---|
| | | Latency | BLEU | Latency | BLEU | Latency | BLEU |
| HAT (SPOS) | 1 | 96.39 | 38.94 | 176.44 | 39.26 | 187.53 | 39.16 |
| HAT (SPOS) | 2 | 98.91 | 38.96 | 159.87 | 39.20 | 192.11 | 39.09 |
| HAT (SPOS) | 3 | 100.15 | 38.96 | 158.67 | 39.24 | 189.53 | 39.16 |
| Layer-wise MoS | 1 | 99.42 | 39.34 | 158.68 | 40.29 | 205.55 | 41.24 |
| Layer-wise MoS | 2 | 99.60 | 39.32 | 156.48 | 40.29 | 209.80 | 41.13 |
| Layer-wise MoS | 3 | 119.65 | 39.32 | 163.17 | 40.36 | 208.52 | 41.18 |
| Neuron-wise MoS | 1 | 97.63 | 39.55 | 200.17 | 40.02 | 184.09 | 41.04 |
| Neuron-wise MoS | 2 | 100.46 | 39.55 | 155.96 | 40.04 | 188.87 | 41.15 |
| Neuron-wise MoS | 3 | 100.47 | 39.57 | 157.26 | 40.04 | 190.40 | 41.17 |

Table 12: Stability of the evolutionary search w.r.t. different seeds on the WMT'14 En-Fr task. Search quality is measured in terms of latency and sampled (direct) supernet performance (BLEU) of the models in the pareto front.

### A.4.6 EVOLUTIONARY SEARCH - STABILITY

We study the initialization effects on the stability of the pareto front outputted by the evolutionary search for different supernets. Table 12 displays sampled (direct) BLEU and latency of the models in the pareto front for different seeds on the WMT'14 En-Fr task. The differences in the latency and BLEU across seeds are mostly marginal. This result highlights that the pareto front outputted by the evolutionary search is largely stable for all the supernet variants.

### A.4.7 IMPACT OF DIFFERENT ROUTER FUNCTION

Table 13 displays the impact of varying the number of hidden layers in the router function for neuron-wise MoS on the WMT'14 En-De task. Two hidden layers provide the right amount of router capacity, while adding more hidden layers results in steady performance drop.

### A.4.8 IMPACT OF INCREASING THE NUMBER OF EXPERT WEIGHTS 'M'

Table 14 displays the impact of increasing the number of expert weights 'm' for the WMT'14 En-Fr task, where the architecture for all the supernets is the top architecture from the pareto front of HAT for the latency constraint of 200 ms. Under the standard training budget ($40K$ steps for MT), the

| # layers in router function | BLEU (↑) |
|---|---|
| 2-layer | **26.61** |
| 3-layer | 26.14 |
| 4-layer | 26.12 |

Table 13: Validation BLEU of different router functions for neuron-wise MoS on the WMT'14 En-De task.

| Supernet | m | BLEU (↑) | Supernet GPU Memory (↓) |
|---|---|---|---|
| HAT | - | 39.13 | 11.4 GB |
| Layer-wise MoS | 2 | **40.55** | 15.9 GB |
| Layer-wise MoS | 4 | 40.33 | 16.1 GB |

Table 14: Impact of increasing the number of expert weights 'm' for the WMT'14 En-Fr task. The architecture is the top model from the pareto front of HAT for the latency constraint of 200 ms.

| Supernet | BLEU (↑) | SacreBLEU (↑) |
|---|---|---|
| HAT | 26.25 | 25.68 |
| Layer-wise MoS | 27.31 | 26.7 |
| Neuron-wise MoS | **27.59** | **27.0** |

Table 15: Performance of supernet as measured by BLEU and SacreBLEU for the latency constraint of 150 ms on the WMT'14 En-De task.

performance of layer-wise MoS does not seem to improve by increasing 'm' from 2 to 4. Increasing 'm' introduces too many parameters, which might necessitate a significant increase in the training budget (e.g., 2 times more training steps than the standard training budget). For fair comparison with existing literature, we use the standard training budget for all the experiments. We will investigate the full potential of the proposed supernets by combining larger training budget (e.g., $\geq 200K$ steps) and larger number of expert weights (e.g., $\geq 16$ expert weights) in future work.

### A.4.9 SACREBLEU VS. BLEU

We use the standard BLEU (Papineni et al., 2002) to quantify the performance of supernet following HAT for a fair comparison. In Table 15, we also experiment with SacreBLEU (Post, 2018), where the similar trend of MoS yielding better performance for a given latency constraint holds true.

### A.4.10 BREAKDOWN OF THE OVERALL TIME SAVINGS

Table 16 shows the breakdown of the overall time savings of MoS supernets versus HAT for computing pareto front for the WMT'14 En-De task. The latency constraints include 100 ms, 150 ms, 200 ms. MoS have an overall GPU hours savings of at least 20% w.r.t. HAT, thanks to significant savings in additional training time (45%-51%).

### A.4.11 CODEBASE

We share the codebase in the supplementary material, which can be used to reproduce all the results in this paper. For both BERT and machine translation evaluation benchmarks, we add a README file that contains the following instructions: (i) environment setup (e.g., software dependencies), (ii) data download, (iii) supernet training, (iv) search, and (v) subnet retraining.

| Supernet | Overall Time (↓) | Supernet Training Time (↓) | Search Time (↓) | Additional Training Time (↓) |
|---|---|---|---|---|
| HAT | 508 hours | **248 hours** | **3.7 hours** | 256 hours |
| Layer-wise MoS | 407 hours (20%) | 262 hours (-5.6%) | 4.5 hours (-21.6%) | 140 hours (45.3%) |
| Neuron-wise MoS | **394 hours (22%)** | 266 hours (-7.3%) | 4.3 hours (-16.2%) | **124 hours (51.6%)** |

Table 16: Breakdown of the overall time savings of MoS supernets vs. HAT for computing pareto front (latency constraints: 100 ms, 150 ms, 200 ms) for the WMT'14 En-De task. Overall time (measured as single NVIDIA V100 hours) includes supernet training time, search time, and additional training time for the optimal architectures. Savings in parentheses.

