# OpenReview forum: "Mixture-of-Supernets: Improving Weight-Sharing Supernet Training with Architecture-Routed Mixture-of-Experts"
_ICLR.cc/2024/Conference — Submitted to ICLR 2024_

### Official Review · Reviewer_GCC7 · 2023-10-25

**Soundness:** 2 fair
**Presentation:** 3 good
**Contribution:** 2 fair
**Rating:** 3
**Confidence:** 4

**Summary:**

The presented paper proposes using MoE's in supernets and shows significant gains for machine translation and language modelling.

**Strengths:**

- Successfully applies MoE supernets to both BERT and MT models demonstrating significant metric improvements
- The work is written in a well structured manner making it easy to follow

**Weaknesses:**

---
### Weaknesses

- **[major]**: As far as I can tell, many of the main contributions are already published in [AutoMoE: Heterogeneous Mixture-of-Experts with Adaptive Computation for Efficient Neural Machine Translation](https://aclanthology.org/2023.findings-acl.580) (Jawahar et al., Findings 2023) for machine translation and the presented work would need to outline how it differs from previous work besides applying it for different tasks.
- **[major]**: Despite WMT'14 being commonly used in the literature, they are now way overhauled in the broader machine translation literature and should be replaced by more recent test sets to put the results into the context of recent research, see **[2]** or **[3]**.
- **[major]**: The presented work should follow the broader machine translation standard to report their evaluation scores using `sacrebleu` and provide the corresponding hash that was used for generating the scores. This will ensure that scores are reproducible and do not vary across papers by up to 1.8 BLEU points due to varying tokenization and normalization, see **[2]**, **[4]**. This should replace the metrics reported in the main paper and not only live in the Appendix.
- **[major]**: While BLEU is still commonly used, there are now better metrics that correlate more closely with human judgement **[3]**, specifically I'd additionally report chrF and COMET scores.
- **[major]**: The machine translation experiments are missing a compute-matched dense and mixture of experts baseline trained from scratch without NAS to make results comparable.


---
### Minor Comments & Typos

- p.2: "Typically, [the] weight-sharing supernet"

---
### Missing References

- **[1]**: [AutoMoE: Heterogeneous Mixture-of-Experts with Adaptive Computation for Efficient Neural Machine Translation](https://aclanthology.org/2023.findings-acl.580) (Jawahar et al., Findings 2023)
- **[2]**: [Non-Autoregressive Machine Translation: It’s Not as Fast as it Seems](https://aclanthology.org/2022.naacl-main.129) (Helcl et al., NAACL 2022)
- **[3]**: [Results of WMT22 Metrics Shared Task: Stop Using BLEU – Neural Metrics Are Better and More Robust](https://aclanthology.org/2022.wmt-1.2) (Freitag et al., WMT 2022)
- **[4]**: [A Call for Clarity in Reporting BLEU Scores](https://aclanthology.org/W18-6319) (Post, WMT 2018)

**Questions:**

- What are top-$k$ in the router set to?

---

> ### Author Response · Authors · 2023-11-18
> **Initial Response**
>
> Thanks for the careful review and for the helpful comments. We respond to quoted comments below:
> * **Weakness 1** Compared to AutoMoE, our work, Mixture-of-Supernets, have entirely different goals, applications of mixture-of-experts, router specifications, as explained below:
>   1. **Goals** Given a search space of dense and mixture-of-expert models, the goal of the AutoMoE framework is to search for high-performing model architectures that satisfy user-defined efficiency constraints. The final architectures can be dense or mixture-of-expert models. On the other hand, given a search space of dense models only, the goal of the Mixture-of-Supernets framework is to search for high-performing dense model architectures that satisfy user-defined efficiency constraints. The final architecture can be a dense model only.
>   2. **Applications of mixture-of-experts** The main application of mixture-of-experts idea by the AutoMoE framework is to augment the standard NAS search space of dense models with mixture-of-experts models. To this end, the AutoMoE framework modifies the standard weight sharing supernet to support weight generation for mixture-of-expert models. On the other hand, the Mixture-of-Supernets framework uses the mixture-of-expert design to: (i) increase the capacity of standard weight sharing supernet and (ii) customize weights for each architecture. Post training, the expert weights are collapsed to create a single weight for the dense architecture.
>   3. **Router specifications** The router underlying the AutoMoE framework takes token embedding as input, outputs a probability distribution over experts, and passes token embedding to top-k experts. On the other, the router underlying the Mixture-of-Supernets framework takes architecture embedding as input, outputs a probability distribution over experts (layer-wise MoS) / neurons (neuron-wise MoS), uses the probability distribution to combine ALL the expert weights into a single weight, and passes token embedding to the single weight.
> * **The machine translation experiments are missing a compute-matched dense and mixture of experts baseline trained from scratch without NAS to make results comparable.** For compute-matched mixture of expert baseline, prior work such as AutoMoE has already shown “training from scratch without NAS” baseline (e.g., random search) clearly underperforms NAS baselines such as HAT (e.g., on the WMT’14 En-De task, random search underperforms HAT by 0.9 BLEU points for the same latency constraint of 600ms).
> * **What are top-k in the router set to?** The number of experts (‘m’) is set to 2 for all the experiments. Note that the router accepts the architecture embedding as input, outputs a probability distribution over experts (layer-wise MoS) / neurons (neuron-wise MoS), merges ALL the experts based on the probability distribution to create a single weight, and passes the token embedding to the single weight. If we have misunderstood your question, can you please clarify?
>
> We hope we have answered some of your questions to your satisfaction. We will follow up with another response to address your remaining questions about sacreBLEU, chrF, COMET and compute-matched dense baseline. Please let us know if there are any further questions that we can address during the interactive rebuttal period.

---

### Official Review · Reviewer_yeUq · 2023-11-05

**Soundness:** 2 fair
**Presentation:** 2 fair
**Contribution:** 3 good
**Rating:** 5
**Confidence:** 3

**Summary:**

The authors propose a novel formulation to improve weight sharing methods in Neural Architecture Search. To improve the expressivity of the model, the authors propose to use Mixtures of Experts (MoE). The MoE blocks are responsible for dynamically generating the weights of the activated subnetwork architecture. In the manuscript, the authors convert BERT to a supernet and use their MoE formulation to improve the performance of the final architecture derived from supernet training, by partially decoupling the weights of the subnetworks. Results shown by their model that SOTA performance is possible across various NLP tasks.

**Strengths:**

1. Results of the method show significant speedup versus baseline and non-marginal improvement in the final model accuracy

2. The method to use Mixtures of Experts (MoE) as a way to perform weight disentanglement is an interesting and novel idea and is backed up by solid quantitative results.

**Weaknesses:**

1. The decision to use MoE as a method for weight disentanglement is not fully justified. A better explanation and introduction is necessary.

**Questions:**

1. In the current MoE formulation the weights for a given architecture are formed of the linear combination of m (2) expert. The combination of experts is handled by the router. Given that all operations appear to be linear, can the router not directly generate the weights?

2. The use of the experts allows the amount of weight sharing between candidate linear operations to be tempered. One architecture may use one set of weights and another can use a completely separate set, or a linear combination of the expert weights. Setting the weights to be completely decoupled (disentangled) would revert the approach to the traditional weight sharing approaches such as in ENAS. Please could the authors explain why a completely disentangled ablation is not performed (m=number of layers). Also can the authors provde an ablation for m=1, i.e. weights fully entangled?

3. The focus of the paper is on removing the need for from-scratch training of the final architecture. However the purpose of training the supernet is generally to find the optimal architecture, not to find optimal weights. Would stopping the supernet training earlier save more time?

4. Explanation of neuron-wise MoS needs a better explanation. Elements of its explanation should be in the earlier section on layer-wise MoS, such as the introduction of m value as the number of experts. Making Fig 1 less busy would help with this.

5. Can the authors explain why the final architecture performs better for having being trained via their method? Is it because the architecture found is better or because the final model's performance is improved by having a better weight initialisation?

---

> ### Author Response · Authors · 2023-11-18
> **Response**
>
> Thanks for the careful review and for the helpful comments. We respond to quoted comments below:
> * **Weakness 1** Thanks for the suggestion. MoE is an established modeling technique to increase capacity of a model and specialize weights to handle a particular set of inputs well. Our method extends this general notion such that the capacity of supernet can be increased and supernets can also specialize weights to handle a particular set of architectures well. As discussed in Section 3.1, MoE can increase the capacity of supernet and naturally does not alter the subnetwork architecture compared to other ways such as adding hidden layers or hidden neurons, which is desirable for our supernet formulation. We will add this text in the revision.
> * **In the current MoE formulation the weights for a given architecture are formed of the linear combination of m (2) expert. The combination of experts is handled by the router. Given that all operations appear to be linear, can the router not directly generate the weights?** If a router is tasked to generate the weights, the number of learnable parameters increases by a large magnitude, which can significantly increase the optimization difficulty. For instance, if we need to construct a weight matrix of 768 x 768, the architecture embedding size is 10, the router hidden dimension is 64, the router contains two projection matrices (10 x 64) and (64 x (768 x 768)), which amounts to 37M learnable parameters for a single weight matrix only. On the other hand, if the number of experts is 2, the router underlying layer-wise mixture-of-supernets contains two projection matrices (10 x 64) and (64 x 2), which amounts to 768 parameters only.
> * **could the authors explain why a completely disentangled ablation is not performed (m=number of layers).** 'm' is the number of experts and is specific to a weight matrix in the original model. For instance, if m=2, our methods represent each weight matrix in the model using two expert weights. We want to understand why m=number of layers is suggested. Can you please provide more context to your question?
> * **can the authors provde an ablation for m=1, i.e. weights fully entangled?** When m=1 (dense supernet model), mixture-of-supernets generalize to traditional weight sharing supernet (e.g., HAT in MT), which has sub-optimal performance and high retraining time (e.g., mixture-of-supernets reduce the retraining cost by 60% for WMT'14 En-De task as shown in Table 6).
> * **The focus of the paper is on removing the need for from-scratch training of the final architecture. However the purpose of training the supernet is generally to find the optimal architecture, not to find optimal weights. Would stopping the supernet training earlier save more time?** Stopping the supernet training earlier can lead to unreliable performance estimates for search and high retraining time for the final architecture. If we've misunderstood your question, can you please clarify?
> * **Explanation of neuron-wise MoS needs a better explanation. Elements of its explanation should be in the earlier section on layer-wise MoS, such as the introduction of m value as the number of experts....** In the revised pdf (text in red color), we have added text that defines 'm' in Section 3.2 and connected each subsection in Section 3. Let us know if the new text improves the clarity of the presentation of neuron-wise MoS method.
> * **Can the authors explain why the final architecture performs better for having being trained via their method? Is it because the architecture found is better or because the final model's performance is improved by having a better weight initialisation?** Our mixture-of-supernet formulation produces a high-performing supernet thanks to the increase in the supernet’s capacity and reduction in gradient conflict (see A.1.2 for empirical analysis of gradient conflict). Such supernet can enhance the quality and the efficiency of final architecture identified by the search algorithm (e.g., 1.6, 0.2, 10.6, 1.0, 0.2 avg. GLUE points improvements for BERT models of 5M, 10M, 26.8M, 30M, 50M parameters respectively (see Table 3), at least 1 BLEU point improvements (except for WMT’14 En-Fr, 100ms) across different latency constraints and WMT datasets for MT). Also, such a supernet can provide better weight initialization for obtaining the optimal weights for the final architecture and reduce the retraining cost significantly (e.g., the retraining cost of the final model is 0 for BERT evaluation as shown in Table 3 and minimal compared to SOTA NAS framework for MT evaluation as shown in Table 6).
>
> We hope we have answered your questions to your satisfaction. Please let us know if there are any further questions that we can address during the interactive rebuttal period.

---

### Official Review · Reviewer_gRzs · 2023-11-05

**Soundness:** 2 fair
**Presentation:** 2 fair
**Contribution:** 2 fair
**Rating:** 5
**Confidence:** 3

**Summary:**

The paper proposes a mixture-of-supernets approach for Neural Architecture Search in NLP that uses a mixture-of-experts to improve upon the weight customization of subnetworks. This method aims to decrease retraining times and improve training efficiency. Preliminary results suggest that it may achieve better latency-BLEU trade-offs for machine translation models and more memory-efficient BERT models compared to some current methods. The study indicates potential for reducing retraining while maintaining acceptable performance levels.

**Strengths:**

The author introduce a novel method in the form of mixture-of-supernets, blending the mixture-of-experts method to refine the weight-sharing mechanism, which is a novel step in neural architecture search.

The proposed approach is designed to cut down on the need for retraining, which could save significant time and computational resources.
Experimental results show that the method could lead to better-performing machine translation models and more efficient BERT models, indicating an improvement over existing techniques.

**Weaknesses:**

The paper presents some areas for potential improvement:

1.  The study concentrates on smaller networks, which may not face the challenges of larger models where memory and computational resources are more critical. To fully evaluate the method's applicability, incorporating a broader range of network sizes and model families (e.g., BERT, T5, GPT) would be beneficial for assessing its scalability and generalizability.

2. The absence of FLOPs (Floating Point Operations Per Second) as a performance metric is noticeable. Including FLOPs would provide a more complete comparison with other methods. Additionally, the exclusion of the STS-B task, a single regression task in the GLUE benchmark, is not justified; its inclusion could enhance the validation of task generalization.

3. Although efficiency is purported to be a key advantage of the proposed method, this is not thoroughly discussed in the main body of the paper. The supplementary comparison of memory efficiency between HAT and the proposed method indicates a significant memory efficiency gap. Moreover, the improvements in task performance are relatively marginal  and efficiency drop compared to other baselines is significant, suggesting a trade-off that may undermine the method's relative advantage when considering the marginal performance enhancement.

**Questions:**

How does the proposed method scale when applied to larger network architectures that are more commonly used in current large language model (LLM) tasks?

What impact do Floating Point Operations Per Second (FLOPs) have on the evaluation of the proposed method, and how does this metric correlate with the performance gains reported?

Can the inclusion of the STS-B task from the GLUE benchmark offer additional insights into the generalization capability of the proposed method across various NLP tasks, particularly in regression problems?

To what extent can the proposed method's efficiency be improved to close the gap identified in the supplementary materials, and how does this impact the overall utility of the method?

---

> ### Author Response · Authors · 2023-11-18
> **Response**
>
> Thanks for the careful review and for the helpful comments. We respond to quoted comments below:
> * **Weakness 1** Following the evaluation setup of popular NAS benchmarks for NLP, we explore model sizes (less than 200M parameters) and model families (BERT, MT) due to high computational requirements for supernet training. Our evaluation setups already validate the generalizability of our proposed methods for language model pretraining (BERT) and task-specific training (MT). Extending the evaluation setup to bigger model sizes (e.g., billion parameters) and other model families (T5, GPT) is orthogonal to our main research questions, but will be interesting future directions for our work.
> * **Weakness 2** Thanks for the suggestions.
>   1. For BERT evaluation, we compare against NAS-BERT and AutoDistil in terms of model size and performance only (see Table 3) due to two reasons: (i) the search algorithm is explicitly optimized to improve the model_size-performance tradeoff only and (ii) NAS-BERT and AutoDistil do not report FLOPs in their paper or provide code to compute the cost. For MT evaluation, we compare against HAT on only latency and performance, as the search algorithm is explicitly optimized to improve the latency-performance tradeoff only.
>   2. In our work, we only include the following GLUE tasks: CoLA, MRPC, SST2, QNLI, QQP, RTE, and MNLI to be consistent with the evaluation setup of NAS-BERT and AutoDistil (as mentioned in Section 4.1). NAS-BERT and AutoDistil do not share their models, for us to obtain their STS-B performance for comparison with our methods.
> * **Weakness 3**
>   1. In terms of efficiency, the main advantage of our method is to significantly reduce the resources (e.g., time, steps) for retraining the final architectures for convergence once the search is complete. For BERT evaluation, there is no retraining required for our framework, as shown in Table 2 and 3. For MT evaluation, there is minimal retraining required for our framework, as shown in Table 6. This advantage offsets the increase in the resources (time, memory) for supernet (one-time cost) and search, leading to big savings in the overall time. Due to space constraints, we discuss the breakdown of the overall time savings in the appendix (see Section A.4.10).
>   2. The performance improvements obtained by our methods over baselines in both BERT evaluation and MT evaluation are big (not marginal): e.g., 1.6, 0.2, 10.6, 1.0, 0.2 avg. GLUE points improvements for BERT models of 5M, 10M, 26.8M, 30M, 50M parameters respectively (see Table 3), at least 1 BLEU point improvements (except for WMT’14 En-Fr, 100ms) across different latency constraints and WMT datasets for MT. These strong results are also attested by other reviewers: Reviewer g33E (“strong on language modeling and MT”), Reviewer yeUq (“non-marginal improvement in the final model accuracy”), and Reviewer GCC7 (“BERT and MT models demonstrating significant metric improvements”).
> * **"How does the proposed method scale when applied to larger network architectures that are more commonly used in current large language model (LLM) tasks?"** See Weakness 1 response (above).
> * **"What impact do Floating Point Operations Per Second (FLOPs) have on the evaluation of the proposed method, and how does this metric correlate with the performance gains reported?"** See Weakness 2 (1) response (above).
> * **"Can the inclusion of the STS-B task from the GLUE benchmark offer additional insights into the generalization capability of the proposed method across various NLP tasks, particularly in regression problems?"** See Weakness 2 (2) response (above).
> * **"To what extent can the proposed method's efficiency be improved to close the gap identified in the supplementary materials, and how does this impact the overall utility of the method?"** The proposed method brings in big performance improvements, big overall time savings (including big retraining time savings), at the cost of high supernet GPU memory (see Table 14), supernet training time, and search time (see Table 16). These costs are not very high (e.g., at most 7.3% increase in supernet training time as shown in Table 15), which should not prevent future research in adopting our method especially for big performance improvements and big overall time savings.
>
> We hope we have answered your questions to your satisfaction. Please let us know if there are any further questions that we can address during the interactive rebuttal period.

---

### Official Review · Reviewer_g33E · 2023-11-11

**Soundness:** 2 fair
**Presentation:** 2 fair
**Contribution:** 2 fair
**Rating:** 5
**Confidence:** 3

**Summary:**

The authors leverage MoE techniques to improve upon Supernets, a common framework for NAS. They do experiments on MT and LMs, and show strong results on WMT and GLUE datasets respectively.

**Strengths:**

- Results seem strong on language modeling and MT.
- Leveraging MoE techniques for modeling is intuitive.

**Weaknesses:**

- Paper is hard to follow and took multiple readings to understand the problem setting. Any reader not familiar with NAS area is likely to feel lost. I recommend significantly reworking the problem setup for clarity. This will greatly improve the paper. (Eg. Section 2 is confusing, and extra prose to explain what a supernet is informally and why it is used would be helpful). Connecting each subsection in Section 3 is also needed.
- k-shot NAS is extremely similar to the proposed method, and should not be taxing to reimplement. Given that this is an intuitive baseline even if k-shot NAS didn't exist, I believe this, or something similar (iterative vs joint training) is an important baseline. Without this, it is not clear how effective the proposed method is.
- More than training steps, cost (FLOPS and wall clock time) while training both initial model and final model is a more informative metric than what is displayed in Table 3.

**Questions:**

- When is neuron level NAS useful/practical compared to layer NAS, and vice-versa?

---

> ### Author Response · Authors · 2023-11-18
> **Response**
>
> Thanks for the careful review and for the helpful comments. We respond to quoted comments below:
> * **Weakness 1** In the revised pdf (text in red color), we have added text that introduces supernet informally, usefulness of supernet (see Section 2), and connected each subsection in Section 3. Let us know if the new text improves the clarity.
> * **Weakness 2** Neuron-wise MoS has the key difference to k-shot nas, that the smaller subnets are not a subset of larger ones anymore. Also, reimplementing the k-shot NAS methods (originally defined for convolution architectures for image classification tasks) and adapting the training recipe (e.g., identifying supernet settings, search settings) for BERT/MT tasks is non-trivial. k-shot NAS is not an intuitive baseline as the relevant baselines in NAS for NLP uses standard weight-sharing supernet. For baselines, we primarily use the state-of-the-art frameworks in NAS for BERT (e.g., NAS-BERT, AutoDistil) and NAS for machine translation literature (e.g., HAT).
> * **Weakness 3** Thanks for the suggestion. In Table 3, we compare against NAS-BERT and AutoDistil, both of which do not report FLOPs and wall clock time in their paper or provide code to compute the cost. Nevertheless, our work focuses on reducing the retraining cost of the final model. For our methods, the retraining cost of the final model is 0 for BERT evaluation (as shown in Table 3) and minimal compared to SOTA NAS framework for MT evaluation (as shown in Table 6, which includes wall clock time).
> * **"When is neuron level NAS useful/practical compared to layer NAS, and vice-versa?"** On an average, neuron-wise MoS provides better performance and better retraining efficiency than layer-wise MoS across efficiency constraints (e.g., model size, latency) and NLP tasks (e.g., BERT, MT). We recommend layer-wise MoS when the supernet training resources (e.g., training time, memory) are limited (e.g., the supernet training time of layer-wise MoS is 1.5% faster than that of neuron-wise MoS).
>
> We hope we have answered your questions to your satisfaction. Please let us know if there are any further questions that we can address during the interactive rebuttal period.

---

### Meta-Review · Area_Chair_jWPm · 2023-12-10

**Metareview:**

The authors leverage MoE techniques to improve upon Supernets, a common framework for NAS. They do experiments on MT and LMs, and show strong results on WMT and GLUE datasets respectively.

The reviewers are not convinced by their motivation. In addition, the empirical part is not convincing.

**Justification For Why Not Higher Score:**

n/a

**Justification For Why Not Lower Score:**

n/a

---

### Decision · Program_Chairs · 2024-01-16

Reject